# Multilayer Regulation of *Neisseria meningitidis* NHBA at Physiologically Relevant Temperatures

**DOI:** 10.3390/microorganisms10040834

**Published:** 2022-04-18

**Authors:** Sara Borghi, Ana Antunes, Andreas F. Haag, Marco Spinsanti, Tarcisio Brignoli, Enea Ndoni, Vincenzo Scarlato, Isabel Delany

**Affiliations:** 1Immune Monitoring Laboratory, NYU Langone Health, 550 First Avenue, New York, NY 10016, USA; sara.borghi@nyulangone.org; 2Department of Pathology, NYU Grossman School of Medicine, 550 First Avenue, New York, NY 10016, USA; 3MabDesign, 69007 Lyon, France; ana.sm.antunes@gmail.com; 4School of Medicine, University of St Andrews, North-Haugh, St Andrews KY16 9TF, UK; afh22@st-andrews.ac.uk; 5Institute of Infection, Immunity and Inflammation, University of Glasgow, 120 University Place, Glasgow G12 8TA, UK; 6School of Cellular and Molecular Medicine, University of Bristol, Bristol BS8 1TH, UK; tarcisio.brignoli@gmail.com; 7Lonza Group AG, 4057 Basel, Switzerland; enea.ndoni@gmail.com; 8GlaxoSmithKline (GSK) Vaccines, 53100 Siena, Italy; marco.x.spinsanti@gsk.com; 9Department of Pharmacy and Biotechnology (FaBiT), University of Bologna, 40126 Bologna, Italy; vincenzo.scarlato@unibo.it

**Keywords:** *Neisseria* *meningitidis*, NHBA, thermoregulation, 4CmenB, vaccine

## Abstract

*Neisseria* *meningitidis* colonizes the nasopharynx of humans, and pathogenic strains can disseminate into the bloodstream, causing septicemia and meningitis. NHBA is a surface-exposed lipoprotein expressed by all *N.* *meningitidis* strains in different isoforms. Diverse roles have been reported for NHBA in heparin-mediated serum resistance, biofilm formation, and adherence to host tissues. We determined that temperature controls the expression of NHBA in all strains tested, with increased levels at 30–32 °C compared to 37 °C. Higher NHBA expression at lower temperatures was measurable both at mRNA and protein levels, resulting in higher surface exposure. Detailed molecular analysis indicated that multiple molecular mechanisms are responsible for the thermoregulated NHBA expression. The comparison of mRNA steady-state levels and half-lives at 30 °C and 37 °C demonstrated an increased mRNA stability/translatability at lower temperatures. Protein stability was also impacted, resulting in higher NHBA stability at lower temperatures. Ultimately, increased NHBA expression resulted in higher susceptibility to complement-mediated killing. We propose that NHBA regulation in response to temperature downshift might be physiologically relevant during transmission and the initial step(s) of interaction within the host nasopharynx. Together these data describe the importance of NHBA both as a virulence factor and as a vaccine antigen during neisserial colonization and invasion.

## 1. Introduction

*Neisseria meningitidis* is a strictly human, Gram-negative, commensal diplococcus that asymptomatically colonizes the nasopharynx of 10–35% of healthy individuals [1,2]. However, for reasons not yet fully understood, *N. meningitidis* can cross the epithelial barrier and invade the host, leading to life-threatening infections, especially in infants and adolescents [3,4,5,6,7]. To successfully colonize and invade the host, *N. meningitidis* has developed several mechanisms to monitor multiple characteristics of its surroundings and consequentially control the expression of adhesion molecules [8,9], biofilm formation [10,11], iron acquisition [12,13], metabolism [14,15,16], and immune evasion factors [2,17]. One of these key signals is temperature. The temperature within the upper respiratory tract is affected by the passage of air during respiration of the host, the precise anatomical location, and the presence of local inflammation [18,19]. Such factors can result in an overall variability of temperature in this niche, ranging from 25.3 ± 2.1 °C in the nasal vestibule to 33.9 ± 1.5 °C in the nasopharynx, generally being several degrees below the core body temperature [18]. Three meningococcal genes, *cssA*, *lst*, and *fHbp*, which encode factors contributing to capsule biosynthesis, lipopolysaccharide sialylation, and factor H binding protein (fHbp), respectively, show increased expression at 37 °C or above [20], indicating that temperature acts as a fundamental signal to trigger immune evasion. The molecular mechanism underlying this response has been elegantly demonstrated to rely on *cis*-acting RNA thermosensors [20,21,22,23]. Classical RNA thermosensors are typically found in the 5′ untranslated region (5′UTR) of an mRNA molecule and prevent its translation at lower temperatures by sequestering the ribosome binding site sequence through the formation of a stem-loop structure [24,25]. However, RNA thermosensors can also induce protein expression following cold shock. Although cold-induced activation of specific promoters has been implicated in upregulating some cold-shock genes, a major role in cold adaptation is played by post-transcriptional mechanisms, as has been thoroughly described for *Escherichia coli* cold-shock protein A (CspA) [26,27,28,29]. Two different structural conformations of *cspA* mRNA exist as a function of temperature. While at 37 °C, the entire translational initiation region (TIR) is buried within a double-stranded structure, at 10 °C, the alternative conformation imposes structural constraints that expose the ribosomal binding site resulting in higher transcript stability and translatability [30]. Comparative proteome analysis in *N. meningitidis* revealed differential protein expression levels between 32 °C and 37 °C, predominantly affecting the bacterial envelope [31]. Neisserial Heparin Binding Protein (NHBA), NMB1030, and NMB2095 showed the greatest degree of temperature-dependent regulation and were identified as factors contributing to the temperature dependence of autoaggregation, biofilm formation, and cellular adherence [31].

NHBA is a surface-exposed lipoprotein ubiquitous in meningococcal strains of all serogroups [32,33]. Analysis of gene sequences from genetically diverse serogroup B strains revealed the existence of more than 400 distinct peptide variants, which are associated with clonal complexes and sequence types [33,34,35]. In strains belonging to the clonal complex ET-5, such as MC58, the *nhba* gene has a 1467-bp coding sequence which is preceded by a 150-bp Contact Regulatory Element of Neisseria (CREN) involved in the induction of the downstream associated genes upon contact with the target eukaryotic cells [8,36]. The NHBA protein mediates adhesion by binding to heparan sulfate proteoglycans on epithelial cells [37]; it promotes neisserial microcolonies and biofilm formation by binding extracellular DNA, a major component of the biofilm matrix [38], and it contributes to the survival of the non-capsulated *N. meningitidis* in human serum [39,40]. Furthermore, NHBA is one of the major protein antigens of the serogroup B meningococcal vaccine 4CMenB (*Bexsero* (*Bexsero* is a trademark owned by or licensed to the GSK group of companies)) [41] and can induce antigen-specific bactericidal antibodies in both animals and humans [39,42]. Therefore, understanding the mechanisms that regulate the expression of NHBA in circulating meningococci is an important goal to better understand *N. meningitidis* pathogenesis and vaccine-induced responses.

In this study, we explore NHBA expression and regulation in response to temperature changes, and we show that NHBA expression is increased at lower temperatures as a result of a multifaceted regulation, in particular at post-transcriptional and post-translational levels. Finally, we show that increased expression of NHBA correlates with increased surface exposure of the protein and results in higher susceptibility to anti-NHBA antibodies.

## 2. Materials and Methods

### 2.1. Bacterial Strains and Culture Conditions

*N. meningitidis* strains used in this study are listed in Appendix A and were routinely grown on chocolate agar (Biomerieaux, Marcy-l’Étoile, France), Gonococcus (GC) agar (Difco, Franklin Lakes, NJ, USA) supplemented with Kellogg’s supplement I, or on Mueller–Hinton (MH) agar (Difco) at 30 °C or 37 °C as specified for single experiments and 5% CO_2_ overnight. Liquid cultures were grown under the same conditions in GC with Kellogg’s supplement I or in Mueller–Hinton broth (MHB) plus 0.25% glucose. Colonies from overnight growth were used to inoculate 7 mL cultures at an optical density at 600 nm (OD_600_) of ~0.05. The cultures were incubated at 30 °C or 37 °C with shaking until early exponential (OD_600_~0.25), mid-exponential phase (OD_600_~0.5), or as specified in the individual experiments. When required, isopropyl β-D-1-thiogalactopyranoside (IPTG) (Sigma, St. Louis, MO, USA) was added to the culture medium at the indicated final concentrations. Strains were stocked in GC medium with 15% (*v*/*v*) glycerol and were stored at −80 °C. 

*E. coli* DH5-α and Mach-1 (Thermo Fisher Scientific, Waltham, MA, USA) strains were grown in Lysogeny broth (LB), and when required, ampicillin, kanamycin, and chloramphenicol were added to achieve final concentrations of 100, 25, and 10 μg mL^−1^, respectively.

### 2.2. Generation of Plasmids and Neisseria meningitidis Recombinant Strains

DNA manipulations were carried out routinely as described for standard laboratory methods [43]. Lists of oligonucleotides and plasmids used in this study are in Appendix A, respectively. All deletion mutant strains, Δ*nhba*, MC58Δ*nhba*-C*nhba*, and MC58Δ*nhba*-Ptac_*nhba* were obtained as described in [39]. To obtain the pCOM_14aa_mCherry plasmid used to generate the MC58Δ*nhba*-Pwt_mCherry strain, polymerase incomplete primer extension (PIPE) PCR [44] was performed. The coding sequence of the mCherry gene optimized for *Neisseria* (Thermo Fisher Scientific) was amplified with mCherryFw and mCherryRev primers. The pCOM_*nhba* plasmid was used as a template to amplify the backbone with divergent primers: Vector Fw, which maps in the chloramphenicol resistance cassette, and Pwt_14aa Rev allows the amplification of the intergenic region upstream to *nhba*, including its own promoter and the first 14 amino acids of the coding sequence. NmmCh_Nde Fw and NmmCh_Nsi Rev primers were used to amplify the mCherry sequence (Thermo Fisher Scientific). Afterwards, the amplicon was subcloned into pCOM-PInd using the NdeI and NsiI restriction sites included within the sequence. 

The correct nucleotide sequence of each plasmid was confirmed by DNA sequencing. Prior to transformation into *N. meningitidis*, plasmids were linearized by restriction digestion using SpeI (New England Biolabs, Ipswich, MA, USA) following the manufacturer’s instructions. For the transformation of naturally competent *N. meningitidis*, five to ten freshly grown colonies were resuspended in 30 µL of GC medium, 1–10 µg of linearized plasmid DNA was added, and the suspension was spotted onto GC plates, allowed to dry, and incubated for 5–6 h at 37 °C. Transformants were then selected on GC plates containing erythromycin (5 µg mL^−1^), kanamycin (150 µg mL^−1^), or chloramphenicol (5 µg mL^−1^) after overnight incubation at 37 °C. Single colonies were re-streaked on selective media and genomic DNA purified by crude lysis after overnight incubation at 37 °C. Single colonies were resuspended in 100 µL of distilled water, boiled for 10 min, and centrifuged in a benchtop centrifuge for 5 min at 8000× *g*. One to three microliters of the sample were used as a template for PCR analysis to check the correct insertion of the resistance marker by double homologous recombination.

### 2.3. Polyacrylamide Gel Electrophoresis and Western Blotting

*N. meningitidis* colonies from overnight plate cultures were resuspended in PBS to an OD_600_ of 0.5. One milliliter of the suspension was centrifuged for 5 min at 15,000× *g* and the pellet resuspended in 100 µL 2X SDS-PAGE loading buffer (50 mM Tris Cl [pH 6.8], 2.5% (*w*/*v*) SDS, 0.1% (*w*/*v*) bromophenol blue, 10% (*v*/*v*) glycerol, 5% (*v*/*v*) β-mercaptoethanol, 50 mM dithiothreitol [DTT]). For liquid culture, 1 mL of each sample was collected and the concentration normalized in 2X SDS-PAGE loading buffer relative optical density at 600 nm corresponding to the same sample concentration as for plate grown cultures. Samples were denatured at 95 °C for 10 min, and protein extracts were separated by SDS-PAGE on NuPAGE Novex 4–12% Bis-Tris Protein Gels in MES 1X (Thermo Fisher Scientific) and transferred onto nitrocellulose membrane using an iBlot Dry Blotting System (Thermo Fisher Scientific). Membranes were blocked for 2 h at room temperature with PBS with 0.05% (*v*/*v*) Tween 20 (Sigma) and 10% (*w*/*v*) powdered milk (Sigma) and then incubated for 60 min at room temperature with the specific primary antibodies diluted in PBS + 0.05% (*v*/*v*) Tween 20 (Sigma) and 3% (*w*/*v*) powdered milk (Sigma). A horseradish peroxidase-conjugated anti-mouse IgG antibody and the Western Lightning ECL (Perkin Elmer, Waltham, MA, USA) were used according to the manufacturer’s instructions, and the densitometry quantification was performed by using the ImageJ 1.6 software.

### 2.4. Quantitative Real-Time PCR (qRT-PCR) Experiments

Bacterial cultures were grown in 7 mL of the liquid medium to OD_600_ as specified for the individual experiments. Three ml of the culture were then poured onto 3 mL of frozen medium to chill the culture and stop transcriptional changes immediately. Cells were then harvested by centrifugation at 3400× *g* for 10 min. Total RNA was isolated using the RNeasy Mini kit (Qiagen, Hilden, Germany) as described by the manufacturer. The second step of the DNase treatment was performed using RQ1 RNase-free DNase (Promega, Madison, WI, USA) for one hour at 37 °C and purified with the RNeasy Mini kit. RNA was quantified using a Nanodrop 1000 spectrophotometer, and its overall quality was assessed by running samples on a 1% agarose gel. Two μg of total RNAs were reverse-transcribed using random hexamer primers and SuperScript II RT (Thermo Fisher Scientific), following the manufacturer’s instructions. As negative controls, all RNA samples were also incubated without reverse transcriptase.

qRT-PCR was performed in triplicate per sample in 25 μL reaction volumes using Platinum SYBR Green qPCR SuperMix-UDG with Rox (Thermo Fisher Scientific) according to the manufacturer’s instructions and containing 2.5 ng of cDNA, and 0.4 μM of gene-specific primers (Appendix A). Amplification and detection of specific products were performed with an Mx3000P Real-Time PCR system (Agilent Technologies, Santa Clara, CA, USA) using the following cycling conditions: 95 °C for 10 min, followed by 40 cycles of 95 °C for 30 s, 55 °C for 30 s, and 72 °C for 30 s then ending with a dissociation curve analysis. The *16S RNA* gene was used as the endogenous reference control, and the relative transcript change was determined using the 2^−ΔΔCt^ relative quantification method. Two-way ANOVA was used to calculate statistical significance and significance levels are indicated on the respective figures (* *p* < 0.05; ** *p* < 0.01; *** *p* < 0.001).

### 2.5. FACS Analysis

Strains were grown in GC or MHB broth until an OD_600_ of 0.25, collected by centrifugation (8000× *g* for 5 min), incubated for 1 h at room temperature with secondary antibody alone, mouse monoclonal antibodies, or mouse polyclonal sera diluted to specific concentrations indicated in the respective experiments in PBS containing 0.1% (*w*/*v*) BSA. The cells were then incubated for 1 h at room temperature with a secondary rabbit anti-mouse immunoglobulin G (whole molecule) FITC-conjugated (Sigma) and then incubated for 2 h at room temperature in PBS containing 0.5% (*w*/*v*) paraformaldehyde (PFA). After a final washing step, cells were resuspended in 100 µL of PBS, and 7 µL of each sample were plated on MH plates and incubated overnight at 37 °C to confirm the inactivation of the bacteria. All data were collected using a BD FACS CANTO II (Thermo Fisher Scientific) by acquiring 10,000 events, and data were analyzed using the Flow-Jo software (v.8.6, TreeStar Inc., San Francisco, CA, USA).

### 2.6. RNA Stability Assay

The *N. meningitidis* NGH38 wild-type strain was grown at 30 °C or 37 °C until the mid-exponential phase, and active transcription was stopped by adding rifampicin (200 μg mL^-1^). Each culture was kept at 30 °C or 37 °C for 5 min by using a pre-warmed water bath. Three ml of culture were collected at different timepoints, immediately brought to 4 °C by adding to an equal volume of frozen media and processed for total RNA isolation. 

### 2.7. Protein Stability Assay

The *N. meningitidis* MC58 wild-type strain was grown at 30 °C or 37 °C until the mid-exponential phase, and active translation was stopped by adding spectinomycin (150 μg mL^−1^). Each culture was split into two equal parts and incubated at either 30 °C or 37 °C for 2 h. Whole-cell protein extracts were taken at different time points and analyzed by Western blotting.

### 2.8. Serum Bactericidal Assay (SBA)

Serum bactericidal activity against *N. meningitidis* strains was evaluated as previously described [45], with pooled baby rabbit serum (Cederlane) used as the complement source (rSBA). Bacteria were grown in Mueller–Hinton broth plus 0.25% (*w*/*v*) glucose and the indicated concentration of isopropyl β-D-1-thiogalactopyranoside (IPTG) (Sigma) at 37 °C (or 30 °C, as specified in each experiment) with shaking until early exponential phase (OD_600_ of ~0.25) and then diluted in Dulbecco’s saline phosphate buffer (Sigma) with 0.1% (*w*/*v*) glucose and 1% (*w*/*v*) BSA (*Bovine serum albumin*) to approximately 10^5^ CFU mL^−1^. The incubation with 25% of baby rabbit complement with or without polyclonal mouse serum at different dilutions was performed at 37 °C (or 30 °C, as specified in each experiment) with shaking for 60 min. Serum bactericidal titers were defined as the serum dilution resulting in a at least 50% decrease in the CFU mL^−1^ after 60 min of incubation of bacteria with the reaction mixture, compared to the control CFU mL^−1^ at time zero. 

## 3. Results

### 3.1. NHBA Expression and Surface Exposure Are Temperature-Dependent

We investigated how a range of temperatures, which mimic those present at different relevant stages of the meningococcal pathogenesis, might affect NHBA expression levels. Three strains, M11719, 8047, and MC58, representing three different clonal complexes with or without the CREN element, and expressing both long (p3) and short (p20) isoforms of the NHBA protein, were grown overnight on GC agar plates at physiologically relevant temperatures ranging from 28 °C to 40 °C. Western blot analyses on total cell extracts showed that the amount of the NHBA protein in the three strains tested was maximum at 28 °C and 30 °C, and it decreased at 35 °C and higher temperatures (Figure 1a). In contrast, the amount of fHbp was higher at elevated temperatures and decreased with temperature reduction, as previously demonstrated [22], albeit less clearly for M11719, while the expression of Hfq, used as a loading control, remained unaltered at all temperatures. These results indicate that expression levels of the NHBA and fHbp proteins are inversely regulated by temperature changes. To understand whether temperature regulation of NHBA is conserved among different *N. meningitidis* isolates, we expanded the analysis to a broader panel of strains belonging to different clonal complexes, carrying differing variants and long or short isoforms of NHBA. Western blot analyses carried out on total cell extracts from 16 *N. meningitidis* strains (Appendix A) grown until exponential phase at 30 °C and 37 °C revealed thermoregulation of NHBA expression in all strains, being always higher at 30 °C than at 37 °C (Figure 1b). The differences in expression between 30° and 37° shown in Figure 1b varied from little (e.g., strain M11822 and M10713) to large differences (strain M11719 and M11205); however, the degree of difference did not correlate with long or short isoforms, peptide variants or other polymorphisms in the promoter region. Furthermore, the analyses also showed different banding patterns depending on the strain background and NHBA variant/isoform present, likely due to the different extent of NalP processing of the protein. As NHBA is a surface lipoprotein, and the full length and N-terminal processed forms are surface-exposed, we used a flow cytometry assay to verify whether increased expression levels of NHBA might result in increased levels of NHBA exposure on the bacterial cell surface. Indeed, Figure 1c shows that the amount of NHBA protein exposed on the surface is higher at 30 °C compared to 37 °C, thus confirming that there is a correlation between the amount of NHBA expression as measured in total cell extracts and its exposure on the bacterial surface.

### 3.2. NHBA Is Expressed during Active Growth

NHBA is known to be expressed maximally during active replication of the bacterium [46]. To determine the NHBA expression profile during growth at 30 °C and 37 °C, we grew the MC58 strain in liquid culture at both temperatures and harvested samples at various timepoints for both mRNA and protein analysis (Figure 2a). Total cell extracts were used in Western blots to assess NHBA protein expression profiles (Figure 2b), and band intensities were quantified (Figure 2c). At both temperatures, NHBA protein levels were higher during exponential growth and declined when entering the stationary phase. When comparing samples taken at the same growth phase for the different temperatures, NHBA levels were consistently higher at 30 °C compared to 37 °C (5–6-fold, Figure 2c). We used qRT-PCR to investigate the *nhba* transcription profile (Figure 2d). In accordance with the NHBA protein quantification (Figure 2c), the *nhba* transcript was abundant during exponential growth and declined in the stationary phase. However, expression levels were generally higher at 30 °C with respect to 37 °C, with a three-fold difference in the late exponential phase. Interestingly, the protein amount at 30 °C in the stationary phase was high and did not correspond with the transcript levels measured in the same condition, which were almost absent. Therefore, while modest differences in the transcript steady-state levels were measurable between the two temperatures tested, they were much more apparent at the protein levels for all phases of the growth, hence suggesting that rather than being controlled only transcriptionally, NHBA thermoregulation could also be mediated post-transcriptionally.

### 3.3. NHBA Thermoregulation Is Not Driven by the nhba Promoter

To investigate the molecular mechanisms involved in NHBA thermoregulation, an *nhba* deletion mutant (Δ*nhba*) was constructed in the MC58 strain background by replacement of the gene with an erythromycin antibiotic resistance cassette, and in turn, different isogenic complementation mutants were generated (Figure 3a). To test if the genomic context played a role in *nhba* regulation, the entire coding sequence and the upstream intergenic regulatory region were inserted into the NMB1428-NMB1429 genomic locus, generating the C_*nhba* strain. As shown by Western blot analyses (Figure 3b), the C_*nhba* strain showed the same expression pattern and thermoregulation of *nhba* at 30 °C and 37 °C as the MC58 wild-type strain, indicating that *nhba* thermoregulation is independent of the genome context. To determine whether the upstream regulatory elements were required for thermoregulation, we fused the full intergenic region, including the promoter, the CREN sequence, and the initial part of the coding sequence corresponding to the first 14 amino acids to an *mCherry* reporter gene generating the Pwt_*mCherry* strain (Figure 3a). As independent control, we also generated Ptac_*mCherry* strain, a reporter gene carrying the upstream regulatory region of *nhba*, but under the control of the IPTG-inducible promoter (Figure 3a). Expression of mCherry detected by Western blotting of samples collected at either 30 °C or 37 °C (Figure 3c) showed no thermoregulation suggesting that the promoter region on its own cannot account for the observed differences in expression at the two temperatures. To determine whether the combination of the upstream regulatory region without the promoter and the coding sequence of *nhba* were required for thermoregulation, we replaced the MC58 wild-type promoter region with an IPTG-inducible Ptac promoter immediately upstream of the CREN sequence (Figure 3a), generating the Ptac_*nhba* strain. We observed an IPTG dose-dependent increase in *nhba* expression (Figure 3d), and overall NHBA levels at each IPTG concentration were lower at 37 °C compared to 30 °C. Together these results highlight that the promoter *per se* is not driving NHBA thermoregulation; rather, it acts predominantly at the post-transcriptional level.

### 3.4. Temperature Affects nhba mRNA Half-Life

To assess whether *nhba* mRNA stability was affected by the temperature, we evaluated the *nhba* mRNA decay after stopping active transcription by the addition of rifampicin. To obtain higher accuracy and reliability, we used NGH38 as the test strain since overall *nhba* expression levels were higher compared to MC58 (Figure 1b). The culture was grown to mid-log, and RNA was extracted at different timepoints after rifampicin addition. The relative mRNA amount over time was quantified by qRT-PCR, and the mRNA half-life was calculated (Figure 4). As a positive control, we measured the decay of the NMB0838 mRNA, a paralog of the CspA protein, which in *E. coli* is post-transcriptionally thermoregulated [30]. *NMB0838* mRNA showed higher stability at lower temperatures in the NGH38 meningococcal strain. As a negative control, we measured the decay of the *fHbp* mRNA, for which the mechanism of thermoregulation has been described to rely on translatability and not the stability of the transcript [22]. As expected, we did not observe any difference in *fHbp* transcript half-life between the two temperatures. Instead, while the *nhba* transcript was rapidly degraded at 37 °C, its mRNA was considerably more stable at 30 °C, with a half-life (t_1/2_ = 2.08 min; 95% CI = 1.3–3.3 min) double that observed at 37 °C (t_1/2_ = 0.97 min; 95% CI = 0.6–1.6 min) indicating that temperature affects the mRNA half-life of *nhba*.

### 3.5. NHBA Protein Shows Higher Stability at 30 °C Respect to 37 °C

Having determined that NHBA protein levels are strongly affected post-transcriptionally, we wanted to investigate whether altered protein stability at different temperatures could also account for the observed differences. Hence, we grew the MC58 wild-type strain in GC broth at 30 °C and 37 °C to the mid-exponential phase and stopped protein translation by adding spectinomycin. Each culture was then split and incubated at both 30 °C and 37 °C. Samples for whole-cell protein extraction were withdrawn at different timepoints after treatment. NHBA accumulation upon protein translation arrest was analyzed by Western blot with representative results shown in Figure 5. From cultures grown at 30 °C (Figure 5a, upper panel), the amount of NHBA full-length protein appears relatively constant at 30 °C for 45–60 min and significantly decreases after 60–120 min. Whereas, when shifted to 37 °C after treatment, NHBA levels started to decrease already after 30 min and continued to decrease for the remainder of the experiment (Figure 5a, lower panel). Bacteria grown at 37 °C and treated with spectinomycin at 37 °C showed a significant decrease after 30 min (Figure 5b, lower panel), while cultures that had undergone a temperature downshift to 30 °C retained stable NHBA expression levels throughout (Figure 5b, upper panel). Interestingly the 49 kDa processed band also exhibits a significant decrease at 37 °C with respect to 30 °C, similar to the full-length protein. This excludes that the instability of NHBA at higher temperatures is due to higher NalP protease activity and suggests a non-specific temperature-dependent turnover of the full-length and processed protein forms is occurring. Overall, these data suggest that NHBA turnover is driven by temperature, resulting in faster degradation at 37 °C and, accordingly, it appears stable at 30 °C.

### 3.6. NHBA Expression Levels Correlate with Susceptibility to Complement-Mediated Killing by anti-NHBA Antibodies

Since NHBA, similar to fHbp, is a surface-exposed antigen, NHBA expression levels may affect bacterial susceptibility to complement-mediated killing in the presence of anti-NHBA antibodies. We first used the recombinant strain MC58 Ptac_*nhba* (Figure 3a), where NHBA expression is under IPTG-inducible control. Using the increasing concentration of IPTG, we evaluated NHBA expression levels (Figure 6a), surface exposure (Figure 6b), and bactericidal activity of anti-NHBA antibodies (Figure 6c). The relative quantification obtained by densitometry analysis and FACS geometric mean calculation were estimated and, together with rSBA titers, were used to calculate the Pearson correlation coefficients (Figure 6d–f). The analysis showed an almost perfect correlation between NHBA expression and surface exposure (r = 0.993; *p* = 0.0007) and importantly the rSBA titers also correlated with both NHBA expression (r = 0.907; *p* = 0.0931) and with its level of surface exposure (r = 0.973; *p* = 0.0011), confirming the role of NHBA expression levels in the ability of anti-NHBA antibodies to mediate complement-dependent killing. To address whether temperature could also play a role in altering NHBA-mediated susceptibility to anti-NHBA antibody responses, we grew MC58 at both 30 °C and 37 °C and performed the serum bactericidal activity (SBA) assay using anti-NHBA and anti-fHbp antibodies as control. Clear differences in NHBA and fHbp surface antigen expression were observed by Western blot and flow cytometry analysis (Figure 6g), consistent with their inverse temperature regulation. Furthermore, the trends in rSBA titers were in line with the relative antigen expression levels: generally higher rSBA titers were observed for NHBA antiserum at lower temperatures (299 ± 195 at 30 °C vs. 171 ± 74 at 37 °C) while fHbp rSBA titers increased with temperature (5461 ± 2365 at 30 °C vs. 10,926 ± 4735 at 37 °C), albeit that they did not reach statistical significance threshold (*p* < 0.05) (Figure 6h). Altogether these results indicated that under conditions that favor high NHBA expression levels, such as the lower temperature of the nasopharynx niche, *N. meningitidis* might be more susceptible to anti-NHBA antibodies.

## 4. Discussion

Meningococcus has successfully evolved to survive in the human nasopharynx, which is its only natural habitat. Colonization is an essential but considerably challenging process in meningococcal survival and, therefore, a prerequisite for strain carriage as well as for establishing invasive disease [3]. Collectively, the various gene expression strategies employed by *N. meningitidis* constitute a complex symphony of regulatory mechanisms which enable the bacterium to enter the human host, colonize it and escape its defense systems. The orchestration of these expression strategies must ensure that the correct genes are expressed at the appropriate time and in the appropriate location, enabling the coordinated expression of an arsenal of virulence factors in response to external stimuli. Temperature is one of the most relevant stimuli in the nasopharynx, where different gradients are present. The temperature on the mucosal surface of the anterior nares is around 30 °C at the end of inhalation and rises to around 34 °C in the posterior nasopharynx and tonsillar region [18]. Both these sites on the mucosal surface are significantly cooler than the core body temperature of 37 °C, at which the bacterium duplicates during invasive disease. Even slight temperature differences may have a profound impact on the phenotype and proteome of *N. meningitidis* [31]. Adhesive properties of *N. meningitidis* appear stronger at 32 °C compared to 37 °C, and one of the most induced proteins at low temperature is NHBA [31]. NHBA is an important virulence factor for *N. meningitidis*, which is involved in adhesion [37], biofilm formation [10,11] and has been implicated even in serum resistance through its ability to bind heparin [39,40]. 

In this study, we investigated how physiologically relevant temperatures, which mimic the initial colonization stage of pathogenesis, may affect NHBA expression. We show that NHBA expression is at its highest level between 28 °C and 30 °C, also corresponding to higher surface exposure. The regulation in response to temperature is conserved among all the sixteen different strains tested, irrespective of the clonal complex, the NHBA variant and/or the isoform expressed, and the presence or absence of upstream CREN regulatory region, or extent of NalP protease cleavage suggesting that NHBA thermoregulation is a conserved mechanism of major relevance in meningococcal pathogenesis. NHBA expression is growth-phase dependent [46], and here we show that at both 30 °C and 37 °C, the amount of *nhba* transcript is higher at the exponential phase while almost undetectable in the stationary phase, indicating that active transcription occurs only during replicative growth. By contrast, the protein is still abundant even at the stationary phase in cultures grown at 30 °C, while no bands were detectable in extracts from bacteria grown at 37 °C, indicating differences in protein stability between 30 °C and 37 °C, which we confirmed in this study with protein stability evaluation. Taken together, these data indicate that NHBA regulation occurs via both multiple transcriptional and post-translational mechanisms, in response to environmental conditions but also in response to the physiological state of the bacterial cell. Maximal NHBA expression will occur when the bacteria are actively dividing and under temperatures that will be encountered during colonization in the nasopharynx.

Through molecular complementation experiments, we excluded that genomic context played a role in this regulation since introducing the wild-type, gene-including upstream promoter regulatory regions into another genomic locus preserved temperature regulation. Interestingly, fusing the promoter region to an *mCherry* reporter gene did not result in temperature regulation of *mCherry*, and conversely, when replacing the wild-type *nhba* promoter with an IPTG-inducible one, temperature-dependent differences in NHBA protein levels were retained. These experiments indicated that thermoregulation was not dependent on the promoter transcriptional elements. A possible role of σ^E^ in mediating a higher biofilm formation at 32 °C has been suggested [31]. However, a direct role of σ^E^ in the higher expression of *nhba* was excluded. Here we show that there are two distinct post-transcriptional mechanisms of thermoregulation of NHBA protein levels. Firstly, the *nhba* transcript has a shorter half-life at 37 °C than at 30 °C, which is likely to result in lower NHBA expression levels at 37 °C. Although the molecular mechanism behind this observation has yet to be elucidated, it is possible that differences in mRNA conformation might play a role. RNA thermosensors, *trans*-acting RNA, or RNA-binding proteins are different mechanisms through which mRNA stability and translatability can be regulated in response to temperature changes [47]. It is not surprising that strictly human pathogens, incredibly well adapted to their niche reservoir, use such strategies to ensure a particularly rapid and cost-effective response to temperature shifts. Remarkably, RNA thermosensors have been implicated in the thermoregulation of several *N. meningitidis* virulence factors, albeit in promoting expression at 37 °C or higher temperatures. However, RNA conformational changes involved in transcript stability and translatability have also been shown to promote expression at lower temperatures. The RNA thermosensor driving *E. coli cspA* thermoregulation has been thoroughly characterized [30], and more recently, a similar RNA conformational switch has been suggested to drive the expression of *Staphylococcus aureus cidA* [48], *cspB*, and *cspC* [49]. It is possible that a similar mechanism could drive *nhba* mRNA stability and translatability in response to temperature changes. As fusing the native *nhba* promoter (including a sequence corresponding to the first 14 amino acids of NHBA) to the sequence of an *mCherry* reporter gene did not result in significant differences in mCherry protein levels, it appears that other regulatory elements present in the *nhba* coding sequence downstream of the 5′UTR and translational initiation site are required. We hypothesize that low temperatures might promote an RNA conformation within the coding sequence that positively affects *nhba* mRNA stability. However, from the data presented here, we cannot exclude the influence of other post-transcriptional regulation mechanisms, such as *trans*-acting small regulatory RNAs [50] or the involvement of RNA helicase that could directly promote mRNA stabilization and translatability. The discrepancy observed between *nhba* mRNA and protein expression levels in bacteria grown at 30 °C until the stationary phase was reached leads us to hypothesize that further regulation may occur post-translationally. By looking at NHBA protein turnover at 37 °C and 30 °C following inhibition of *de novo* protein translation, we verified that protein stability is different at the two temperatures. In fact, we show that NHBA is turned over rapidly at 37 °C, while its stability is increased significantly following the temperature downshift to 30 °C. The NHBA protein is cleaved specifically by NalP [39], a surface-exposed protease known to process a number of surface antigens of meningococcus when its expression is phase ON [51,52,53]. Interestingly NalP expression was reported to increase at 37 °C with respect to 32 °C [31], suggesting higher NalP specific protease activity may further add specific processing at higher temperatures in addition to the non-specific temperature-dependent turnover of full-length and processed N-terminal NHBA products reported here. Taken together, our results indicate that the overall NHBA expression results from the cumulative effects of multilayer regulation, where both RNA stability and protein turnover play a major role.

NHBA is one of the three major components of the 4CmenB (*Bexsero*) vaccine against serogroup B meningococcus and is a protective antigen able to elicit a robust immune response [39]. Although NHBA is present in all neisserial species, we have shown that its expression is variable among strains and, moreover, is affected by different factors such as growth phase and temperature changes. It is therefore paramount to understand how different NHBA expression levels, either through strain variation or triggered by different growth conditions, affect the bacterium’s susceptibility to anti-NHBA antibodies-mediated killing. The correlate of protection for meningococcal vaccines is the Serum Bactericidal Assay (SBA) which is generally expected to be performed at 37 °C. Using an IPTG-inducible recombinant strain as a model, it was possible to specifically estimate the influence of different NHBA expression levels on bacterial killing during the SBA without introducing pleiotropic alterations caused by different incubation temperatures. We also measured a trend in the higher killing of the meningococcal test strains in SBA with anti-NHBA antibodies when grown at a lower temperature and, inversely, a higher killing with anti-fHbp antibodies at a higher temperature, in line with respective thermoregulated protein expression. While statistical significance was not reached in these experiments, a clearly correlated trend was observed, and we reason that altering the temperature may affect many different processes in the bacterial cell. Confounding pleiotropic effects may make it difficult to determine the impact of specific antigen expression levels on serum bactericidal killing. 

NHBA is conserved among pathogenic *Neisseria* species, and its expression levels theoretically should be higher during colonization where the bacterium is exposed to lower temperatures. The thermoregulation of NHBA is in line with a significant role for this protein in the colonization of the nasopharynx [31,37,38]. The correlation of NHBA expression and killing susceptibility has an important implication: the higher expression of NHBA during the initial steps of transmission/colonization might also result in higher susceptibility to anti-NHBA antibodies in the nasopharyngeal niche. In addition, anti-NHBA antibody responses may be antagonistic towards other functions of NHBA, such as interfering with its adhesive property or its contribution to biofilm. Vaccination against NHBA could supply anti-NHBA antibodies to the colonized mucosa and prevent the initial colonization of the host even before entering the bloodstream. As such, the inclusion of NHBA as a vaccine antigen might prove beneficial in preventing the initial stages of colonization of vaccinated hosts, although direct evidence of antibody-mediated carriage decrease upon vaccination is still lacking.

Taken together, our results underline the importance of NHBA both as a virulence factor and as a vaccine antigen and shed light on the molecular mechanisms that regulate its expression during meningococcal colonization.

## Figures and Tables

**Figure 1 microorganisms-10-00834-f001:**
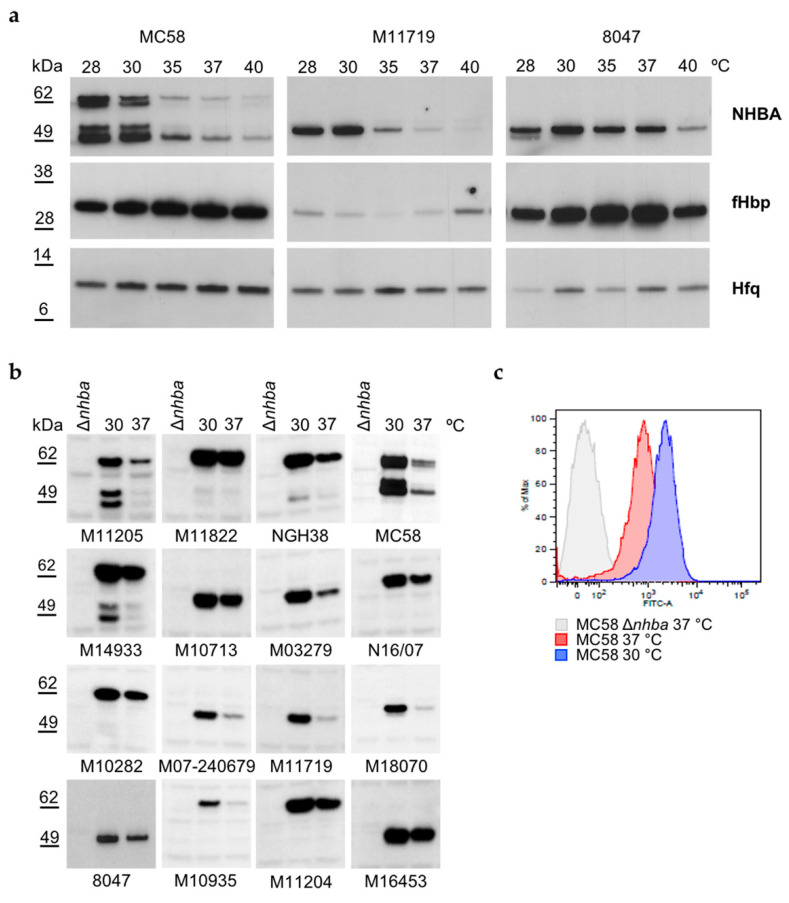
NHBA expression and surface exposure are increased at lower temperatures. (**a**) Serogroup B *N. meningitidis* strains were grown overnight on GC agar plates at the indicated temperatures. Whole cell lysates were prepared and separated by SDS-PAGE prior to Western Blotting. NHBA, fHbp and Hfq proteins were detected using specific mouse-polyclonal antisera. Hfq served as loading control between different samples. In MC58 the full-length protein migrates with an apparent molecular weight of approximately 62 kDa (p3 long isoform), while the other bands right below 62 kDa and at approximately 49 kDa result from processing through NalP bacterial protease [39]. M11719 and 8047 express short isoforms (p20) migrating approximately 50 kDa and do not exhibit NalP processing. (**b**,**c**) The indicated strains were grown in GC broth at 30 °C or 37 °C until OD_600_ 0.25. (**b**) Whole cell lysates were separated by SDS-PAGE and blotted using an anti-NHBA polyclonal antiserum. (**c**) NHBA surface exposure on MC58 and its Δ*nhba* mutant strain at the indicated conditions. Mouse polyclonal antiserum against NHBA was used for flow cytometry analysis.

**Figure 2 microorganisms-10-00834-f002:**
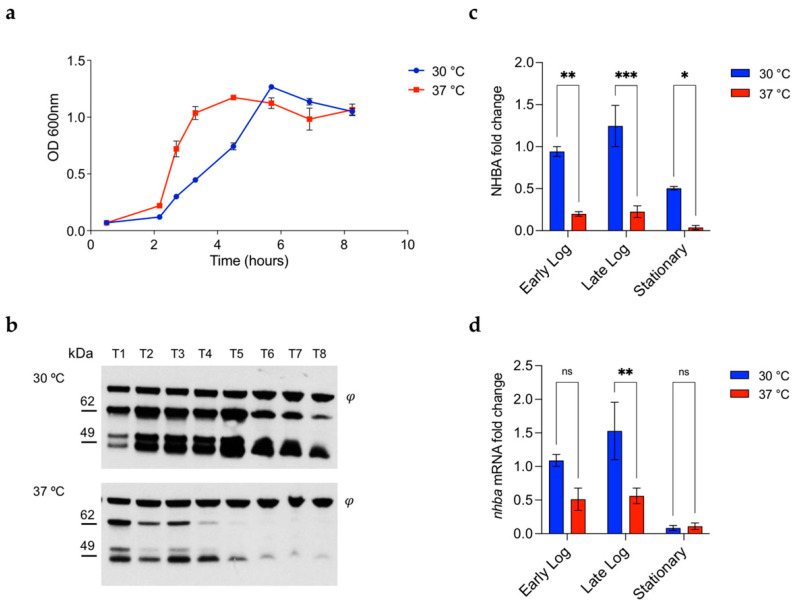
NHBA is highly expressed during the exponential phase and its expression at 30 °C is always higher than at 37 °C. (**a**) Growth profiles of MC58 strain in GC liquid medium at 30 °C (blue) and 37 °C (red). Samples were collected at the indicated consecutive time points (T1-T8) and whole cell lysates were analyzed for NHBA expression by Western blot. (**b**) φ indicates a non-specific band used as loading control and for relative protein quantification. (**c**) Relative protein quantification was calculated using ImageJ software, comparing early logarithmic phase (OD_600_~0.25; T2_37°C_; T3_30°C_), late logarithmic phase (OD_600_~0.85; T3_37°C_; T5_30°C_) and at stationary phase (OD_600_~1.20; T6_37°C_; T7_30°C_) summing the signals of the full length and processed bands of NHBA for each timepoint and normalizing against the non-specific band (φ). (**d**) *nhba* mRNA steady state levels were quantified by qRT-PCR and relative expression levels were determined normalizing to the *16S RNA*. Histograms represent the mean ± SEM from three independent biological replicates and were analyzed by Two-way ANOVA followed by uncorrected Fisher’s LSD multiple comparison test (*** *p* < 0.001; ** *p* < 0.01; * *p* < 0.05; ns: not significant).

**Figure 3 microorganisms-10-00834-f003:**
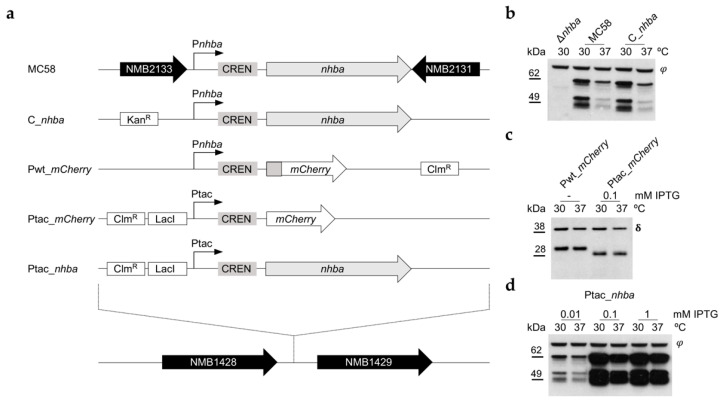
NHBA thermoregulation acts at the post-transcriptional level. (**a**) Schematic representation of *nhba* mutants generated by *ex-locus* complementation. In the MC58 Δ*nhba* strain background different mutants were generated by complementation in the NMB1428-NMB1429 locus. Kan^R^: kanamycin resistance cassette; Clm^R^: chloramphenicol resistance cassette; LacI: LacI repressor gene; Ptac: IPTG inducible promoter; *mCherry*: *mCherry* reporter gene. (**b**–**d**) MC58 wild-type and recombinant strains were grown in GC liquid medium at 30 °C or 37 °C until OD_600_ 0.5 and supplemented with the indicated concentration of IPTG where needed. NHBA and *mCherry* protein expression were assessed by Western blotting using polyclonal mouse antiserum and monoclonal mouse antibody (ab167453, abcam), respectively. The φ and δ symbols indicate non-specific bands used as loading controls.

**Figure 4 microorganisms-10-00834-f004:**
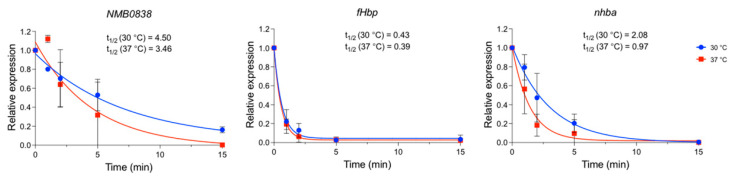
*nhba* transcript has longer half-life at 30 °C. Strain NGH38 was grown in GC broth until OD_600_ 0.5 at the defined temperatures. RNA extracts were prepared at different timepoints after active transcription was stopped by adding rifampicin. *NMB0838*, *fHbp*, and *nhba* mRNA abundance were measured by qRT-PCR and normalized to *16S RNA*. Relative RNA quantity was calculated as 2^−(Ct-Ct0)^. Data represent the mean of three independent biological replicates ± SD. One phase decay analysis for transcripts at 30 °C (blue line) and 37 °C (red line) are represented.

**Figure 5 microorganisms-10-00834-f005:**
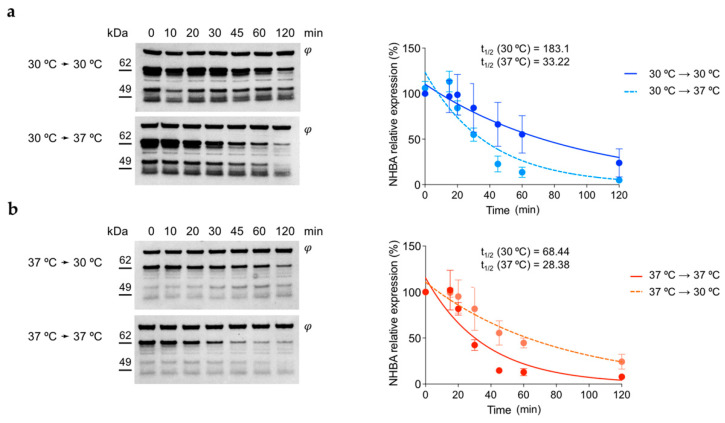
NHBA protein turnover is directly affected by temperature changes. The wild-type strain MC58 was grown in GC broth until OD_600_ 0.5 at (**a**) 30 °C, and (**b**) 37 °C. Active translation was stopped by the addition of spectinomycin, the cultures split into equal volumes and incubated further at the indicated temperatures. Samples for whole cell extracts were collected and prepared at different timepoints thereafter. Protein samples were separated by SDS-PAGE prior to Western blotting. The NHBA band was detected using polyclonal mouse anti-NHBA serum. Relative protein quantification was calculated using ImageJ software, summing the signals of the full length and processed bands of NHBA for each timepoint and -normalizing against the non-specific band (φ) used as loading control, and plotted over time and the half-life quantified for each landing temperature after translation inhibition.

**Figure 6 microorganisms-10-00834-f006:**
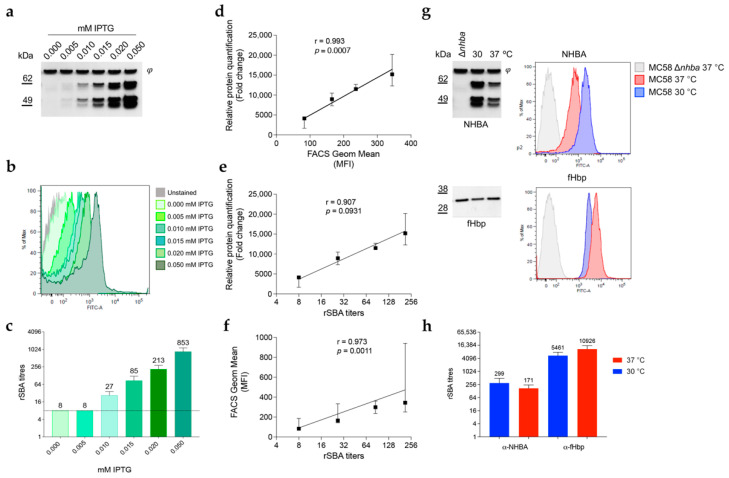
Correlation between NHBA expression, surface exposure and susceptibility to complement-mediated killing by anti-NHBA antibodies. MC58 Ptac_*nhba* strain was grown in MH broth + 0.25% glucose at 37 °C until OD_600_ 0.25. IPTG was added at the indicated final concentrations. Bacteria were collected to perform (**a**) Western blotting, (**b**) FACS analysis and (**c**) serum bactericidal assay. (**d**–**f**) The relative quantification obtained by densitometry analysis of the NHBA-related bands in Western blot and FACS geometric mean calculation were estimated and, together with rSBA titers, were used to calculate the Pearson correlation coefficients. Data represent the median with the error range from three independent biological replicates. Line indicates the linear fit (**d**) or the semilog fit (**e**,**f**). Samples treated with 0.050 mM IPTG or with no IPTG were not taken into account to calculate the Pearson correlation coefficients, as these were found to be out of the linearity range of the assays. (**g**) MC58 wild-type strain was grown at 30 °C or 37 °C in MH broth + 0.25% glucose until OD_600_ 0.25 was reached. Bacteria were collected and expression levels of NHBA and fHbp were determined by Western blotting, while surface exposure was confirmed by flow cytometry using polyclonal antisera. (**h**) Serum bactericidal titers were determined using baby rabbit complement as source of complement factors (rabbit SBA, rSBA). Data are representative of three independent biological replicates.

## Data Availability

Not applicable.

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
