# Peer review of "Multilayer Regulation of Neisseria meningitidis NHBA at Physiologically Relevant Temperatures"

_microorganisms, 2022, doi:10.3390/microorganisms10040834_

Round 1
Reviewer 1 Report
This manuscript by Borghi et al investigated how physiologically relevant temperatures and growth phase may affect NHBA expression, a heparin-binding protein important in meningococcal virulence and one of the recombinant proteins in the meningococcal serogroup B vaccine, Bexsero. The authors showed that NHBA expression and corresponding surface exposure are higher at 30°C than 37°C. The temperature-mediated changes were shown among sixteen different strains tested. An increased RNA stability and higher NHBA protein stability at lower temperatures contributed to the increased NHBA levels. Post-transcriptional and post-translational regulation that required the NHBA coding sequence, not a transcriptional control by the nhba promoter, were responsible for NHBA thermoregulation. Increased NHBA expression resulted in higher susceptibility to complement-mediated killing by antibodies against NHBA. The authors proposed that NHBA regulation in response to temperature downshift might be physiologically relevant during transmission and the initial interactions within the host nasopharynx. While a broad set of experiments nicely demonstrated NHBA thermoregulation, the investigation into the molecular mechanism(s) to elucidate how thermoregulation occurs is rather limited and reduces the overall impact of the study.
Specific comments:
- Page 5, three strains M11719, 8047 and MC58 were examined in greater details. What is the reasoning for selection of these strains?
- The differences in expression between 30C and 37C shown in Fig. 1b varied from little (e.g., strain M11822) to significant (e.g., M11719). What might be the explanation for the variability? Does the degree of difference correlate with promoter SNPs, the presence or absence of CREN or the long/short forms?
- 1, is the processed 49-kD NHBA protein also surface exposed and detected by flow analysis? Are the flow signals total of full-length and processed forms?
- Page 6, the sentence “NHBA protein levels were highest during the exponential growth of the bacterium and they were determined as ~5- and ~6-fold higher at early and late log phase at 30°C rather than 37°C, respectively” is confusing. Please modify.
- Figure 2c, which bands are included in protein quantification? Is it just the full length or the total signals of both 62 & 49 kD bands?
- The Western blots in Fig. 5 should also show the 49kD bands, a protease-processed form, present in strain MC58 examined here. Is the NHBA protein stability due to overall lower protease activity against NHBA at low temp or is there nonspecific turnover of NHBA protein? If due to nonspecific protein turnover, a similar decrease should be detected in the 49-kD form. It would best to evaluate the protein stability with a non-processed full-length NHBA such as the NGH38 strain used for mRNA stability analysis.
- Is the shorter form of NHBA protein also less stable at 37C?
- Since NalP specifically processes NHBA, is NalP expression temperature regulated?
- Is the protein stability difference also true for surface exposed NHBA if examined by flow analysis?
- Page 12. In discussion, it was stated “regulation of expression in response to temperature changes and growth phase are not related and are driven by separate mechanisms”. The reasoning is not clear.
- Table S1, the presence or absence of CREN should be noted for the strains examined.
Author Response
Point by Point response to Reviewer 1 comments
This manuscript by Borghi et al investigated how physiologically relevant temperatures and growth phase may affect NHBA expression, a heparin-binding protein important in meningococcal virulence and one of the recombinant proteins in the meningococcal serogroup B vaccine, Bexsero. The authors showed that NHBA expression and corresponding surface exposure are higher at 30°C than 37°C. The temperature-mediated changes were shown among sixteen different strains tested. An increased RNA stability and higher NHBA protein stability at lower temperatures contributed to the increased NHBA levels. Post-transcriptional and post-translational regulation that required the NHBA coding sequence, not a transcriptional control by the nhba promoter, were responsible for NHBA thermoregulation. Increased NHBA expression resulted in higher susceptibility to complement-mediated killing by antibodies against NHBA. The authors proposed that NHBA regulation in response to temperature downshift might be physiologically relevant during transmission and the initial interactions within the host nasopharynx. While a broad set of experiments nicely demonstrated NHBA thermoregulation, the investigation into the molecular mechanism(s) to elucidate how thermoregulation occurs is rather limited and reduces the overall impact of the study.
Specific comments:
1. Page 5, three strains M11719, 8047 and MC58 were examined in greater details. What is the reasoning for selection of these strains?
Three meningococcal strains (M11719, 8047 and MC58) were selected. These three strains belong to different clonal complexes and summarize the diversity with respect to the presence of the CREN sequence, and long or short isoforms of NHBA. Indeed, the laboratory-adapted reference strain MC58 belongs to clonal complex 32 (the only one presenting the CREN sequence) and expresses a long form of the protein (p3) while strain 8047 and M11719 express short isoforms of NHBA (p20) and belong to clonal complex 11 and 162, respectively. This information has been added to the text (lines 238-239), and details are also reported in Table S1.
2. The differences in expression between 30C and 37C shown in Fig. 1b varied from little (e.g., strain M11822) to significant (e.g., M11719). What might be the explanation for the variability? Does the degree of difference correlate with promoter SNPs, the presence or absence of CREN or the long/short forms?
The differences in expression between 30°C and 37°C shown in Fig1b varied from little (e.g. strain M11822 and M10713) to large differences (e.g. strain M11719 and M11205) however the degree of difference did not correlate with long or short isoforms, peptide variants or other polymorphisms in the promoter region. We stated this in text (lines 253-257).
3. 1, is the processed 49-kD NHBA protein also surface exposed and detected by flow analysis? Are the flow signals total of full-length and processed forms?
The full-length form of NHBA is cleaved by NalP autotransporter protein that is also surface exposed. The C-terminal cleaved fragment is released in the medium (as reported in Serruto et al., 2010) and the 49 kD processed fragment is the N-terminal truncated form of the NHBA lipoprotein. The 49 kDa form is expressed on the surface of the bacterium as a result of processing by NalP. Therefore, the FACS signal is a sum of both full length and processed forms, this has been added to the text (lines 259-262).
4. Page 6, the sentence “NHBA protein levels were highest during the exponential growth of the bacterium and they were determined as ~5- and ~6-fold higher at early and late log phase at 30°C rather than 37°C, respectively” is confusing. Please modify.
We have changed the sentence as follows: “At both temperatures, NHBA protein levels were higher during exponential growth and declined when entering stationary phase. When comparing samples taken at the same growth phase for the different temperatures, NHBA levels were consistently higher at 30C compared to 37C (5-6-fold, Figure 2c).” (lines 289-292)
5. Figure 2c, which bands are included in protein quantification? Is it just the full length or the total signals of both 62 & 49 kD bands?
The protein quantification was made for the full length and processed NHBA-related bands, by summing the signals of the full length and processed bands of NHBA for each timepoint, and normalized against the non-specific band (indicated by φ) and this has been clarified in the legend (lines 316-318).
6. The Western blots in Fig. 5 should also show the 49kD bands, a protease-processed form, present in strain MC58 examined here. Is the NHBA protein stability due to overall lower protease activity against NHBA at low temp or is there nonspecific turnover of NHBA protein? If due to nonspecific protein turnover, a similar decrease should be detected in the 49-kD form. It would best to evaluate the protein stability with a non-processed full-length NHBA such as the NGH38 strain used for mRNA stability analysis.
We have replaced the figure showing the processed bands. Indeed, the 49 kDa fragment is less stable when shifted from 30-37°C but remains unaffected when shifting from 37-30°C fully parallel to the full-length protein. As such, NHBA protein stability does not appear to be due to overall lower protease activity against NHBA at low temperature. We have added this into the text line 408-412.
7. Is the shorter form of NHBA protein also less stable at 37C?
See answer to point 6.
8. Since NalP specifically processes NHBA, is NalP expression temperature regulated?
From proteomic data Lappann et al., NalP appears to be inversely expressed with respect to NHBA with respect to temperature, i.e. downregulated at lower temperature. We have added discussion on this to the text (lines 581-587): ”The NHBA protein is cleaved specifically by NalP [39], a surface exposed protease known to process a number of surface antigens of meningococcus when its expression is phase ON [52-54]. Interestingly NalP expression was reported to increase at 37°C with respect to 32° C [31], suggesting higher NalP specific protease activity may further add specific processing at higher temperatures in addition to the non-specific temperature-dependent turnover of full length and processed N-terminal NHBA products reported here.”
9. Is the protein stability difference also true for surface exposed NHBA if examined by flow analysis?
As we show in Figure 6 d, the measurement of protein expression quantification through densitometry analysis of the NHBA related bands (now clarified in the legend) and FACs analysis of surface expression is extremely highly correlated giving a Pearson coefficient of 0.99 and a p value of 0.0007. Therefore, we expect that the reduction in total NHBA protein associated with cell lysates through temperature sensitive stability absolutely reflects with surface exposed NHBA that may be measured by FACS.
10. Page 12. In discussion, it was stated “regulation of expression in response to temperature changes and growth phase are not related and are driven by separate mechanisms”. The reasoning is not clear.
We have replaced this sentence as follows: “Taken together, these data indicate that NHBA regulation occurs via both multiple transcriptional and posttranslational mechanisms, in response to environmental conditions but also to the physiological state of the bacterial cell. Maximal NHBA expression will occur when the bacteria are actively dividing and under temperatures that will be encountered during colonization in the nasopharynx.” (lines 517-522)
11. Table S1, the presence or absence of CREN should be noted for the strains examined.
We have added the requested information in a new column.
Reviewer 2 Report
In their manusrcipt Borghi et al., describe the regulation of NHBA in N. meningitidis. They investigated the temperature dependent stability of nhba mRNA as well as the NHBA protein as well as its surface exposure and the susceptibility of the bacteria to antibody mediated killing due to the presence of NHBA on the surface. Overall the manuscript is well written and comprehensible. Nevertheless, I would like to address a few points that might help to improve the manuscript.
Materials and Methods
Section „Generation of plasmids …”
“Generation of all deletion mutants… was obtained as described…”
- Either “Generation… was performed” or “All deletion mutants … were obtained…”
“Afterwards, the amplicon was subcloned…”
- Name the exact restriction sites used
Section “Polyacrylamide gel…”
“… resuspended in 100 µl in 2X SDS-PAGE…”
- Remove the “in” after 100 µl
Figure 1:
- At least in part (a) the intensity of the major band relative to the loading control should be added.
- The loading control for strains M11719 and 8047 is missing
- At least one exemplary gel with the marker used should be shown (even a supplementary file would be fine).
- Why is the NHBA band in strain MC58 so much larger than in the other two strains. Please explain.
Figure 2a:
- A fitted line is not allowed for growth curves. If a line is to be shown between the points, it may only connect the individual points.
- The X-axis should be “time [h]”.
Figure 2b:
- As in Fig. 1a a quantification relative to the loading control should be added.
Figure 2 – legend:
- (c) Please add the time points (T1-T8) that were used as representatives for early log, late log and stationary phase
- Move the sentence “The Ï• symbol…” to the description of (c).
Section 3.5:
- The complicated sentence “Each culture was then split…” should be split into two sentences.
Figure 5:
- Show the whole gels. Especially the part with the 49 kDa band.
- Calculate the protein half lives like the RNA half lives shown in Fig. 4. Maybe the diagrams should have a logarithmic Y-axis, as this would lead to a linear trendline that is easier to follow.
Discussion:
- The authors suggest that NHBA could be a good candidate for a vaccine. As due to the described thermoregulation NHBA is only found on the surface of the bacteria in larger quantities as long as they are outside the body, it is unclear to me to what extent they are then still a meaningful target for antibodies that are formed as a result of a vaccination. Please explain.
Author Response
Point by Point response to Reviewer 2 comments
In their manuscript Borghi et al., describe the regulation of NHBA in N. meningitidis. They investigated the temperature dependent stability of nhba mRNA as well as the NHBA protein as well as its surface exposure and the susceptibility of the bacteria to antibody mediated killing due to the presence of NHBA on the surface. Overall the manuscript is well written and comprehensible. Nevertheless, I would like to address a few points that might help to improve the manuscript.
Materials and Methods
1. Section „Generation of plasmids …”
“Generation of all deletion mutants… was obtained as described…”
- Either “Generation… was performed” or “All deletion mutants … were obtained…”
“Afterwards, the amplicon was subcloned…”
This has been modified as follow: ‘All deletion mutant strains, Δnhba, MC58Δnhba-Cnhba and MC58Δnhba-Ptac_nhba were obtained as described in’. Please see lines 118-119.
2. Name the exact restriction sites used
This has been added ‘Afterwards, the amplicon was subcloned into pCOM-PInd using the NdeI and NsiI restriction sites included within the sequence. Please see line 129.
3. Section “Polyacrylamide gel…”
“… resuspended in 100 µl in 2X SDS-PAGE…”
- Remove the “in” after 100 µl
This has been modified accordingly.
4. Figure 1:
- At least in part (a) the intensity of the major band relative to the loading control should be added.
- The loading control for strains M11719 and 8047 is missing
- At least one exemplary gel with the marker used should be shown (even a supplementary file would be fine).
- Why is the NHBA band in strain MC58 so much larger than in the other two strains. Please explain.
We have modified Figure 1 accordingly, adding loading control and fHBP staining from each strain, and adding the marker, (as per uploaded original full Western blot scans). NHBA exists in two principal isoforms (long and short due to a 60 amino acid insertion in the N terminus of long alleles). MC58 expresses the long NHBA isoform (p3) while the other two strains express the short isoform (p20). NHBA is also subject to processing by the NalP protease, when the nalP gene is phase ON, resulting in multiple NHBA bands in the NalP-positive MC58 strain while only a single band in the other two strains which do not exhibit NalP processing. We have added these details to the legend. Table S1 summarizes all the relevant characteristics for the natural isolates used in this study.
5. Figure 2a:
- A fitted line is not allowed for growth curves. If a line is to be shown between the points, it may only connect the individual points.
- The X-axis should be “time [h]”.
We modified and substituted a new Figure 2a according to the reviewer’s comment.
6. Figure 2b:
- As in Fig. 1a a quantification relative to the loading control should be added.
Figure 2b contains a non-specific band indicated by φ and Figure 2c contains a quantification for the timepoints of major interest and for which we have mRNA steady state levels: the quantification has been performed by summing the signals of the full length and processed bands of NHBA for each timepoint, and normalized against the non-specific band and this has been now clarified in the legend. We have modified the figure legend to identify the relevant timepoints, see point 7.
7. Figure 2 – legend:
(c) Please add the time points (T1-T8) that were used as representatives for early log, late log and stationary phase.
We have indicated the respective timepoints in the figure legend to clarify the relevant timepoints.
Line 305-311 '(b). φ indicates a non-specific band used as loading control and for relative protein quantification. (c) Relative protein quantification was calculated using ImageJ software, comparing early logarithmic phase (OD600 ∼ 0.25; T237°C; T330°C), late logarithmic phase (OD600 ∼ 0.85; T337°C; T530°C) and at stationary phase (OD600 ∼ 1.20; T637°C; T730°C) summing the signals of the full length and processed bands of NHBA for each timepoint and normalizing against the non-specific band (φ).'
8. Move the sentence “The Ï• symbol…” to the description of (c).
This has been modified according to the reviewer’s suggestion. Please see lines 316-318.
9. Section 3.5:
- The complicated sentence “Each culture was then split…” should be split into two sentences.
This has been modified accordingly. Please see lines 404-407.
10. Figure 5:
- Show the whole gels. Especially the part with the 49 kDa band.
- Calculate the protein half lives like the RNA half lives shown in Fig. 4. Maybe the diagrams should have a logarithmic Y-axis, as this would lead to a linear trendline that is easier to follow.
We have substituted Figure 5 with a new one included a larger section of the Western blots and have added an extra panel with the quantification of protein half-lives. Indeed, the 49 kDa fragment is less stable when shifted from 30-37°C but remains unaffected when shifting from 37-30°C fully parallel to the full-length protein. To be consistent with our mRNA half-lives graph, we have retained a linear y-axis.
11. Discussion:
The authors suggest that NHBA could be a good candidate for a vaccine. As due to the described thermoregulation NHBA is only found on the surface of the bacteria in larger quantities as long as they are outside the body, it is unclear to me to what extent they are then still a meaningful target for antibodies that are formed as a result of a vaccination. Please explain.
NHBA is thought to be involved in the initial colonization of the host in the nasopharyngeal niche. Here, temperatures are lower than in the blood stream leading to higher NHBA expression. Vaccination against NHBA could supply anti-NHBA antibodies to the colonized mucosa and prevent the initial colonization of the host even before entering the bloodstream. We have attempted to clarify this in the text. Line 591-593
Round 2
Reviewer 2 Report
Even though I would have preferred a logarithmic Y-axis for the presentation of the half-lives (one could use logarithmic axes in both Fig. 4 and Fig. 5), the authors' responses to my comments are satisfactory. I therefore see no reason not to publish the paper in its present form.